**Data Availability Statement:** All relevant data are within the manuscript's Supporting information files.

**Funding:** AKA was the recipient of a Discovery Foundation Rural Fellowship Award in the

# An analysis of obstetric practices and outcomes in a deep rural district hospital in South Africa

**Adam Konrad Asghar** *, **Thandaza Cyril Nkabinde, Mergan Naidoo**

Department of Family Medicine, University of KwaZulu-Natal, Durban, KwaZulu-Natal, South Africa

* adam.asghar@gmail.com

## Abstract

### Background

Internationally, there has been a focus on ensuring that Caesarean deliveries are performed only when indicated, to ensure the best outcome for mother and baby. In South Africa, despite a variety of health system interventions, maternal and perinatal mortality remain unacceptably high.

### Objectives

To describe and compare the clinical outcomes related to the mode of delivery, for patients managed at rural primary healthcare level.

### Methods

This retrospective cross-sectional observational analytical study was conducted at a deep rural district hospital in northern KwaZulu-Natal, South Africa. Maternity Case Records and Caesarean delivery audit tools from 2018 were reviewed.

### Results

In total, 634 files were retrieved. The Caesarean delivery rate in the sample was 30.8% (193 of 634 deliveries), and according to the Robson classification, groups 5 and 1 were the biggest contributors to Caesarean delivery. All Caesarean deliveries were deemed to have been medically indicated. As compared to those whose delivery was normal vaginal, the odds of having post-partum haemorrhage were 25 times higher, and the odds of having any complication were three times higher, if a mother delivered by Caesarean (p<0.001). In neonates who were delivered by Caesarean, the odds of being admitted to nursery were four times higher than those delivered vaginally (p<0.001).

### Conclusion

Showing a significantly higher risk of maternal and neonatal complications, this study validated Caesarean delivery at rural primary care as a potentially dangerous undertaking, for which adequate precautions should be taken. There is a need for interventions targeting

Individual category, in relation to this research (REF: 040511; https://www.discovery.co.za/marketing/discovery-foundation-site/index.html). The funders had no role in study design, data collection and analysis, decision to publish, or preparation of the manuscript.

rural healthcare in South Africa, to ensure that obstetric services are offered to patients in as safe a manner as possible in this environment.

## Introduction

Selection of the mode of obstetric delivery can be a significant determinant of the maternal and perinatal outcomes of a pregnancy, irrespective of the antenatal course [1]. The features of mother-foetus pairs who would benefit from a particular mode of delivery (MOD) are well-established, with some regional variations in practice [2]. However, application of these practices is not consistent, especially in resource-constrained environments, with consequent failure to improve outcomes [3, 4]. It is known that Caesarean delivery (CD) is a potentially life-saving obstetric intervention, but one which has many potential risks, as compared to vaginal delivery [5]. Internationally, there has been a focus on ensuring that CDs are performed only when indicated, and on increasing access to CD for populations that currently have limited access [6].

The latest reports released by the National Committee for Confidential Enquiries into Maternal Deaths ('Saving Mothers'), and the National Perinatal Morbidity And Mortality Committee ('Saving Babies'), show that maternal and perinatal mortality remain unacceptably high. Caesarean delivery contributes to maternal deaths through haemorrhage, anaesthetic complications, CD-associated sepsis and thromboembolism in the index pregnancy, and makes for higher-risk deliveries in the future. Conversely, failure to offer CD timeously, contributes to maternal deaths through pregnancy-related sepsis, haemorrhage and hypertension, and perinatal deaths through birth asphyxia, antepartum haemorrhage, hypertension and birth trauma [7, 8].

The proportion of births occurring by CD as opposed to vaginal delivery is on the rise in South Africa, but this has been accompanied by an increase in maternal mortality related to bleeding during or after CD (BLDACD)–the national Caesarean delivery rate (CDR) increased from 12.7% in 2001/2 to 26.2% in 2015/16 (% of deliveries in facility), whilst the case-fatality rate (CFR) for BLDACD was ∼25 per 100 000 CDs in 2008 and has risen to ∼30 in 2016. Consequently, there has been greater focus at policy level on making CD a safe intrapartum practice [7, 9, 10]. Although the reports from the ministerial committees are derived from in-depth analysis of deaths throughout the country, they provide generalised conclusions. What has not been done recently is to analyse a variety of outcomes specifically at rural primary healthcare level, where up to a third of South African mother-foetus pairs will have their deliveries managed [11].

The purpose of this observational analytical study was to describe and compare the maternal and perinatal outcomes related to the obstetric intra-partum practices, for pregnant women who delivered at a deep rural district hospital in northern KwaZulu-Natal, South Africa in 2018.

## Methods

### Study setting

The study took place in a rural district hospital, located in the poorest quintile of South African districts. The hospital is ranked as one of the 15 most rural (out of 255) district hospitals in the country [11]. The labour ward and operating theatre conduct approximately 150 deliveries monthly. Twenty-four percent of deliveries in 2017 were by CD. The antenatal clinic and

labour ward are run on a midwife model of care, with doctors managing higher-risk patients and admitted antenatal/postnatal patients.

## Study population

The study population included all pregnant women who delivered at the facility in 2018, according to the labour ward register.

## Exclusion criteria

Any delivery not conducted within the labour ward/operating theatre was excluded, as were deliveries of neonates with a birth weight of less than 1 000 g and of stillbirths diagnosed before onset of labour. Lastly, patients whose Maternity Case Records (MCRs) were not found were excluded from the sample.

## Design

This was a cross-sectional retrospective observational analytical study.

## Data collection

In early 2020, hospital records were reviewed by the principal investigator, and data extracted from these sources using a tool which was validated in a pilot study (unpublished). The pilot study was conducted at another district hospital in KwaZulu-Natal, with a sample size of 79 (>10% of the intended sample size for the main study). Content and face validity was established after the pilot study data was analysed, and the research objectives achieved. The main data source was individual MCRs; other sources were also reviewed (inpatient files for admitted neonates, labour ward delivery registers, operating theatre records, facility CD audit tools, minutes from monthly perinatal morbidity/mortality meetings).

## Sampling

The sample size for comparing the maternal outcomes and perinatal outcomes against the modes of delivery (NVD and CD) was calculated using G*Power software (version 3.1 [2020]; HHU, Düsseldorf, Germany). Using a 95% confidence level, power of 0.99, margin of error of 5% and an allocation ration of 1:3.57 (reflecting the actual proportions in each arm of the study population), the estimated sample size was 555. This was determined using the difference of proportions of post-partum haemorrhage (PPH) in each arm from the literature [12]. Systematic random probability sampling of the labour ward delivery register was used to select MCRs.

## Analysis

Data were recorded on an Excel spreadsheet (version 16 [2020]; Microsoft Corp., Redmond, WA, USA) and analysed using SPSS (version 27 [2020]; IBM Corp., Armonk, NY, USA). For maternal outcomes and risk factors, data were analysed at the level of mother. Each neonate was treated as an independent observation, thus for neonatal outcomes, the data were analysed at the level of neonate. Chi-squared and Fisher's exact tests were used to examine associations between categorical variables. Independent t-tests were used to compare means between two groups. Magnitude of association was measured using odds ratios. The level of statistical significance was set at <0.05.

## Ethical considerations

Ethical approval was obtained from the University of KwaZulu-Natal Biomedical Research Ethics Committee (Ref: BREC/00000101/2019), and the KwaZulu-Natal Department of Health's Health Research Committee (Ref: KZ_201909_026). Permission to conduct the study was obtained from the facility and district management.

## Results

### Study setting

The labour ward had three normal delivery beds, one high-care bed, and an emergency blood fridge (in which a small number of packed red cell units were kept). The operating theatre was approximately 90 metres away, and the blood bank over 170 km by road.

During the study period, in addition to the operational manager (an advanced midwife and a Master Trainer in Essential Steps in The Management of Obstetric Emergencies—ESMOE), the maternity service had a monthly median of 20 (range 14–23) registered nurses, some of whom were community-service registered nurses and midwifery students. Nursing staff were distributed between areas and shifts to provide 24-hour cover. Some shifts lacked advanced midwifery cover (total of four advanced midwives), and approximately half of the nursing staff had less than two years of midwifery-specific experience.

The hospital had a median monthly pool of 15 non-specialist doctors, in addition to the medical manager. One or two doctors covered the maternity clinic/ward during the day, and out-of-hours services for the whole hospital were provided by three doctors per shift. Of the doctors, four were foreign-qualified, with very limited obstetric experience, and four were community-service medical officers. The cohort of doctors contained five ESMOE Master Trainers. All doctors were competent in spinal anaesthesia, five in general anaesthesia, and twelve in CD. Since 2015, the medical team regularly audited CDs (analysis conducted by the medical team on the morning of the next working day after the CD was performed). Normal vaginal deliveries (NVDs) were not routinely audited, unless there was an adverse maternal or neonatal outcome.

### Clinical characteristics

The number of women who delivered at the facility in 2018 was 1 621. The initial sample size was 803 (increased beyond the planned sample size due to anticipated difficulties in retrieving MCRs), but four patients were further excluded after sampling, as they were found (with access to the MCR) to fulfil the exclusion criterion of 'stillbirths diagnosed before onset of labour'; of these 799, 634 MCRs were found. Thus n = 634 for deliveries. For neonatal outcomes, n = 649. The characteristics of deliveries are summarised in Table 1, and of neonates in Table 2. Birth-weight centiles were determined using Intergrowth charts [13].

The clinical characteristics of the parturients and neonates were very similar except for maternal weight, number of foetuses (higher maternal weight and multiple pregnancy associated with CD) and gestational age (with a higher gestational age associated with CD).

### Obstetric practices

Looking at obstetric practices, the CDR was 30.8% (195 of 634 deliveries). Of the vaginal deliveries (439) only 1 was an assisted (vacuum) delivery. The modes of delivery according to Robson group are summarised in Fig 1, and Table 3 [14].

In parturients who had CD, the Lucas delivery urgency classes are summarised in Fig 2 and Table 4 [15]. No CD performed was classified as Lucas I.

**Table 1. Clinical characteristics of deliveries.**

| | | NVD | CD | p-value |
|---|---|---|---|---|
| Age | 10–14 | 3 | 1 | |
| | 15–19 | 126 | 42 | |
| | 20–34 | 263 | 127 | |
| | >34 | 47 | 25 | |
| | Mean (SD) | 24.5 (6.9) | 25.6 (6.6) | 0.078 |
| Weight (kg) | Missing | 3 | 1 | |
| | Mean (SD) | 73.5 (14.7) | 76.5 (15.0) | **0.021** |
| Pre-delivery Haemoglobin (g/dl) | <7 | 0 | 0 | |
| | 7–9.9 | 84 | 39 | |
| | 10–10.9 | 115 | 40 | |
| | ≥11 | 235 | 116 | |
| | Missing | 5 | 0 | |
| | Mean (SD) | 11.1 (1.3) | 11.2 (1.4) | 0.453 |
| HIV status | Negative | 282 | 128 | 0.733 |
| | Positive | 157 | 67 | |
| Parity | 0 | 213 | 77 | |
| | 1–4 | 215 | 113 | |
| | >4 | 11 | 5 | |
| | Mean (SD) | 1.0 (1.3) | 1.2 (1.3) | 0.191 |
| Number of foetuses | Single | 434 | 185 | **0.002** |
| | Multiple | 5 | 10 | |
| Leading presentation/lie | Cephalic | 436 | 186 | |
| | Breech | 3 | 8 | |
| | Other | 0 | 1 | |

NVD, normal vaginal delivery; CD, Caesarean delivery; HIV, human immunodeficiency virus.

Two hundred and seventy-eight women (valid percentage 44.2%) were anaemic (defined as a haemoglobin <11 g/dl) pre-delivery.

Adolescents (defined as a maternal age of 10–19 years) accounted for 172 (27.1%) of deliveries, 155 (53.4%) of nulliaparas (total 290 –see Table 3 below, Robson groups 1, 2a, 2b), and 43 (22.1%) of all CDs. The adolescent-specific CDR was 25%.

Of the Lucas II class CDs, the median decision to delivery interval (DDI) was 90 minutes, with a range of 52–287 minutes. There was no difference (p = 0.308) in DDI, if Lucas class II CDs took place during working hours or outside of working hours.

Primary mode of anaesthesia for CD was spinal in 193 (99.0%) cases. Seven (4.0%) of 193 spinal anaesthetics needed conversion to general anaesthesia due to inadequate spinal (three), high spinal (one), or a surgical complication (three).

Looking at the auditing information, only 119 (61.0%) CDs were audited. Of these, all CDs were deemed to have been medically indicated. The WHO CD safety checklist was fully completed in 171 (87.7%) CDs, and partially completed in 11 (5.6%) CDs.

Comparing parturients whose labour started spontaneously with those whose labour was induced, it was found that there was no association with MOD (p = 0.352).

## Outcomes

A variety of maternal and neonatal outcomes are detailed according to MOD in Tables 5 and 6. Post-partum haemorrhage was defined as blood loss exceeding 500ml in NVD, and 1000ml in CD.

**Table 2. Clinical characteristics of neonates.**

| | | NVD | CD | p-value |
|---|---|---|---|---|
| Sex | Female | 217 | 105 | |
| | Male | 227 | 100 | |
| Birthweight (g) | 1000–2499 | 39 | 20 | |
| | 2500–3999 | 396 | 175 | |
| | ≥4000 | 9 | 10 | |
| | Mean (SD) | 3098 (472) | 3155 (571) | 0.209 |
| Gestational age (completed weeks) | <32 | 1 | 0 | |
| | 32–33 | 1 | 3 | |
| | 34–36 | 38 | 10 | |
| | 37–38 | 175 | 68 | |
| | 39–40 | 175 | 99 | |
| | 41 | 45 | 22 | |
| | >41 | 9 | 3 | |
| | Mean (SD) | 38.6 (1.8) | 38.9 (1.6) | **0.029** |
| Birthweight centile | Lowest (<10%) | 28 | 22 | |
| | Middle (10–90%) | 360 | 154 | |
| | Upper (>90%) | 56 | 29 | |
| | Mean (SD) | 58.6 (27.5) | 57.5 (30.2) | 0.663 |
| HIV status | Negative | 139 | 60 | |
| | Positive | 1 | 0 | |
| | Indeterminate | 1 | 0 | |
| | Missing | 16 | 7 | |

NVD, normal vaginal delivery; CD, Caesarean delivery; HIV, human immunodeficiency virus.

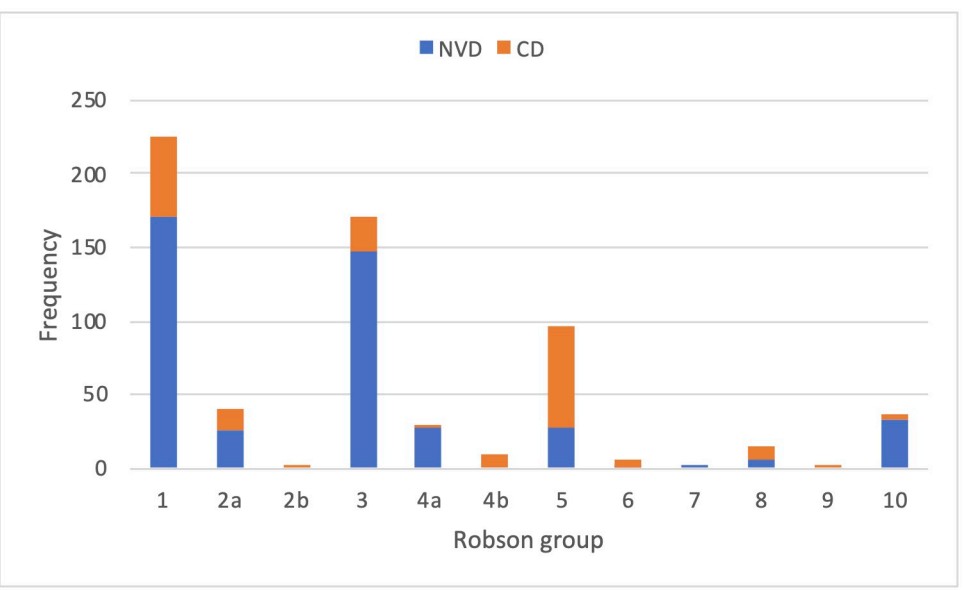

**Fig 1. Mode of delivery counts according to Robson groups.**

**Table 3. Robson groups—Definitions and frequencies.**

| Robson group | Definition | NVD | CD |
|---|---|---|---|
| 1 | Nulliparous; single, cephalic, ≥ 37 weeks, spontaneous labour | 171 | 55 |
| 2A | Nulliparous; single, cephalic, ≥ 37 weeks, induced labour | 26 | 14 |
| 2B | Nulliparous; single, cephalic, ≥ 37 weeks, pre-labour CD | 0 | 2 |
| 3 | Parous, without uterine scar; single, cephalic, ≥ 37 weeks, spontaneous labour | 148 | 22 |
| 4A | Parous, without uterine scar; single, cephalic, ≥ 37 weeks, induced labour | 27 | 3 |
| 4B | Parous, without uterine scar; single, cephalic, ≥ 37 weeks, pre-labour CD | 0 | 10 |
| 5 | Parous, with uterine scar; single, cephalic, ≥ 37 weeks | 28 | 69 |
| 6 | Nulliparous; single, breech | 1 | 4 |
| 7 | Parous with or without uterine scar; single, breech | 1 | 0 |
| 8 | With or without uterine scar; multiple | 5 | 10 |
| 9 | With or without uterine scar; single, oblique/transverse | 0 | 1 |
| 10 | With or without uterine scar; single, cephalic, <37 weeks | 32 | 5 |

NVD, normal vaginal delivery; CD, Caesarean delivery.

It is noted that CD was associated with an increased likelihood of bleeding, infection and interventions related to these complications. Thirty-one CDs (15.9%) had associated anaesthetic complications reported as per Fig 3.

CD in the index pregnancy was associated with having had one previous CD. Of the 82 mothers who had a history of previous CD x 1, 50 were deemed eligible for a trial of vaginal birth (amenable and with no medical contraindications to vaginal birth after CD). Twenty-nine delivered by NVD, giving a rate of successful vaginal birth after one CD of 58%.

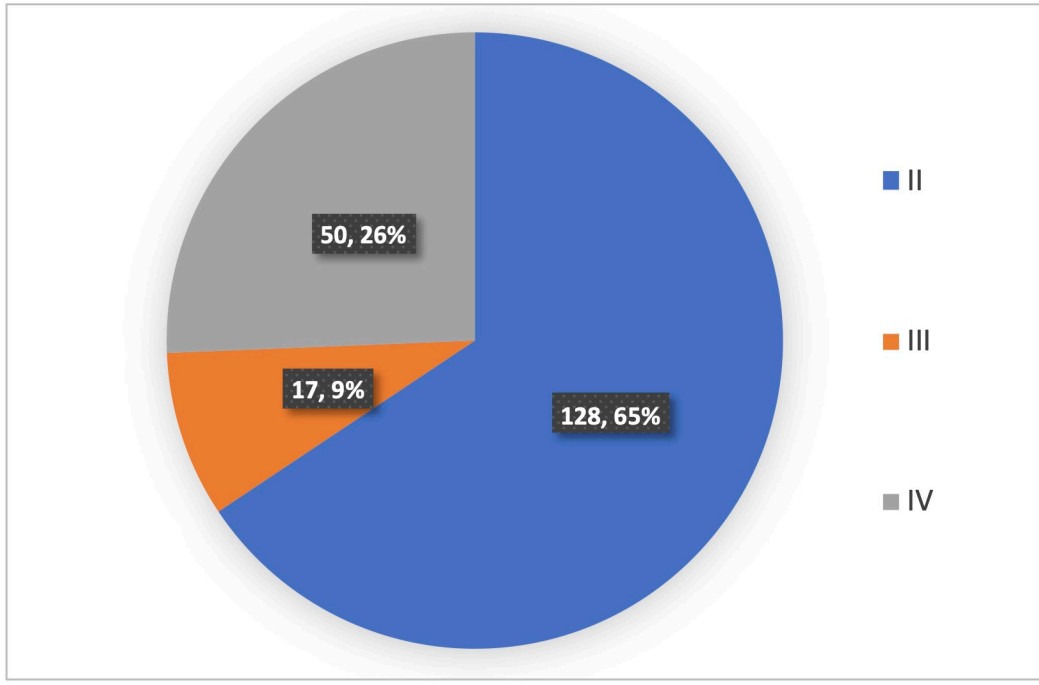

**Fig 2. Urgency of CDs performed according to Lucas class.**

**Table 4. Lucas classification of urgency of CD.**

| Lucas class | Definition |
|---|---|
| I | Immediate threat to life of woman or foetus |
| II | Maternal or foetal compromise which is not immediately life-threatening |
| III | Needing early delivery but no maternal or foetal compromise |
| IV | At a time to suit the woman and maternity services |

A total of six babies born had a 'low Apgar score' of <7 at 5 minutes of life, which gave a rate of 9.2 per 1 000 live births. Four babies died after admission to nursery, which equated to a perinatal mortality rate of 6.2 per 1 000 live births (this figure excluding all stillbirths that were diagnosed before the onset of labour). No statistically significant association was found between MOD and neonatal Apgar score. Neonates who were born by CD were more likely to be admitted to nursery after birth, but there was no association between MOD and the admitted neonate being transferred to a higher level of care or dying.

A secondary analysis was conducted to determine if CD indication and urgency were related to complication rates. No statistically significant associations were found.

## Discussion

The main findings of this study, which sought to describe and analyse obstetric intra-partum practices and the maternal/perinatal outcomes at a deep rural district-level hospital in South Africa, are as follows:

**Table 5. Maternal outcomes.**

| Maternal characteristic | | NVD | CD | Odds ratio | p-value |
|---|---|---|---|---|---|
| Disposition outcomes | Discharged | 438 | 194 | | |
| | Transferred out | 1 | 1 | [a] | 0.521 |
| | Died | 0 | 0 | | |
| Duration of post-partum stay | Median (days) | 1 | 3 | | |
| | % staying longer than median | 13.7 | 16.9 | 1.3(0.8–2.0) | 0.285 |
| Managed at appropriate level of care[b] | Yes | 425 | 189 | | |
| | No | 14 | 6 | 1.0 (0.4–2.7) | 0.941 |
| Bleeding | Missing | 23 | 4 | | |
| | Median (ml) | 200 | 700 | | |
| | PPH | 5 | 45 | **25.3 (9.9–65.1)** | **<0.001** |
| Infection | Puerperal infection | 2 | 5 | **5.8 (1.1–29.9)** | **0.038** |
| Interventions | Blood transfusion required in 7 days before/after delivery | 4 | 14 | **8.4 (2.7–25.9)** | **<0.001** |
| | Non-pneumatic anti-shock garment use | 0 | 1 | [a] | 0.308 |
| | Post-partum laparotomy | 0 | 4 | [a] | **0.009** |
| | Post-partum ventilation | 0 | 1 | [a] | 0.308 |
| Overall maternal complications[c] | | 66 | 59 | **3.1 (2.1–4.6)** | **<0.001** |
| Previous CD x 1 | | 29 | 53 | **6.6 (4.0–10.8)** | **<0.001** |

NVD, normal vaginal delivery; CD, Caesarean delivery; PPH, post-partum haemorrhage.

[a]No statistical analysis performed due to small number of observations.

[b]Assessed in light of provincial obstetric referral criteria [16].

[c]Composite of transferred out, prolonged stay, PPH, puerperal infection and anaesthetic complication.

**Table 6. Neonatal outcomes.**

| Neonatal characteristic | | | NVD | CD | Odds ratio | p-value |
|---|---|---|---|---|---|---|
| Birth outcome | | Stillborn | 0 | 0 | | |
| | | Liveborn, not admitted | 413 | 157 | | |
| | | Liveborn, admitted | 31 | 48 | **4.1 (2.5–6.6)** | **<0.001** |
| Apgar score | | Median score at 5 minutes | 10 | 10 | | 0.080 |
| | | Apgar score <7 at 5 minutes (n) | 3 | 3 | | |
| Birth trauma | | None | 440 | 174 | | |
| | | Cephalhaematoma | 1 | 1 | | [a] |
| | | Erb's palsy | 1 | 0 | | |
| | | Missing | 2 | 30 | | |
| Admission to nursery | Disposition outcome | Discharged | 27 | 46 | | |
| | | Transferred out | 1 | 0 | [a] | 0.078 |
| | | Died | 3 | 1 | [a] | |
| | Duration of stay | Median (days) | 4 | 4 | | 0.431 |
| Overall neonatal complications[b] | | | 30 | 44 | **4.0 (2.4–6.6)** | **<0.001**[b] |

NVD, normal vaginal delivery; CD, Caesarean delivery.

[a] no statistical analysis performed due to small number of observations.

[b] Composite of nursery admission (which includes those transferred out and those who died), Apgar score <7 at 5 minutes, and birth trauma.

- In-facility CDR was 30.8%

- The two Robson groups which contained the highest absolute number of CDs were 5 and 1, respectively

- CD is more likely to be associated with maternal complication (specifically PPH, puerperal infection and anaesthetic complication), and neonatal admission.

As such, this study offers a recent insight into the state of obstetric services in a rural district-level hospital.

## Study setting

The Saving Mothers reports note that maternal mortality related to CD is highest in rural provinces, and that the CFR for CD relative to that for vaginal delivery is 1.4 times higher at primary healthcare level than the national relative CFRs for MOD. The health system has attempted to address such inequities, with a variety of initiatives designed to minimise peri- and post-partum risks to the patients being cared for in the perinatal sphere, including ESMOE, Minimum Standards for Safe Caesarean Delivery and Helping Babies Breathe [17]. The staff at the study setting were familiar and compliant with these programmes.

South Africa's rural areas are home to 43.6% of the country's population, but are served by only 12% of the country's doctors and 19% of its nurses [18]. Such disparities in human resources between rural and urban contexts would be expected to affect practices and auditing standards e.g. decision-making and timing may be influenced by the fact that the doctors expected to perform a CD in a rural district-level hospital are the same doctors who are staffing the casualty department. Regardless of this, the median DDI of 90 minutes for Lucas II CDs does not compare favourably with the international audit standard of 75 minutes; this delay may reflect a large number of variables, which this study did not address [19].

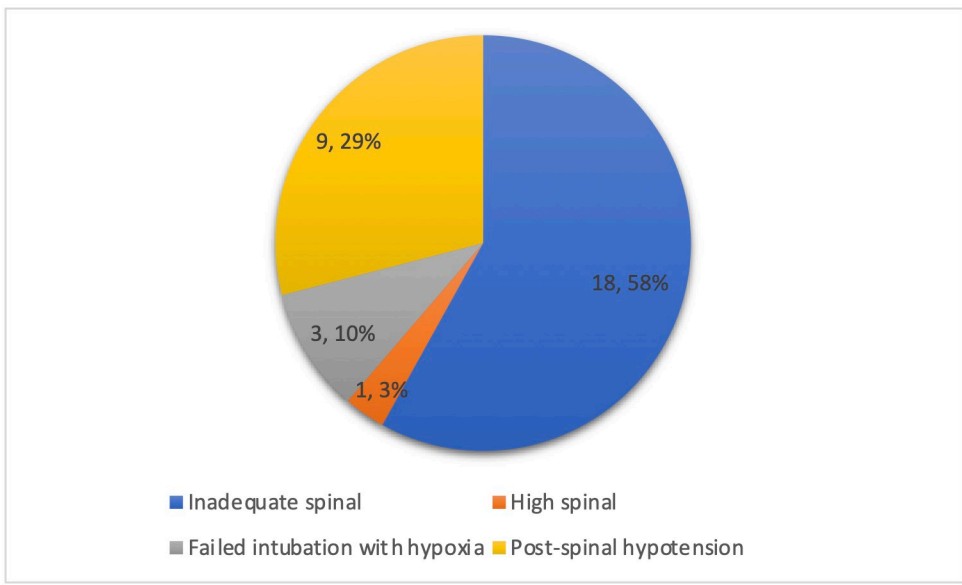

**Fig 3. Frequency of anaesthetic complications reported.**

## Clinical characteristics

A large proportion of women (44.2%) were anaemic pre-delivery. Direct comparisons to other studies should not be made, as the haemoglobin measurement was not done at a uniform time point; the 'latest' test may have been taken weeks prior to delivery, whereas other studies have focussed on anaemia at booking and/or in a different setting [20]. However, this is a significant descriptive finding, when considering that anaemia is a contributory cause of maternal death [21], and that the blood bank is located over two hours away from the facility by road. Given this distance between the facility and the blood bank, it can be postulated that the transfusion practices were conservative; eighteen patients (2.8%) were given a blood transfusion in the seven days prior to and following delivery.

The association found between MOD and gestational age is likely to be clinically insignificant and this over-estimation of precision may be explained by the inequality of sample sizes.

## Obstetric practices

Global data from Molina et al. [22] suggest that the optimal CDR in relation to maternal and neonatal mortality is approximately 19 CDs per 100 live births, which is higher than the previously accepted WHO recommendation of 10–15 CDs per 100 live births [23].

Whilst the figure in our study of 30.8% seems high in relation to the international literature (even more so when adjusted to CDs per 100 live births– 31.6), without considering the setting and the outcomes achieved, comparison of crude rates is not informative [24].

Focusing on local practices, a study profiling CDs done in a South African urban district hospital in 2015 found a rate of 32% [25], whilst Gaunt [26] audited vacuum deliveries occurring in 2014 in a deep rural district-level in the Eastern Cape—within 319 deliveries, he found a CDR of 17.8%, and an assisted delivery rate of 7.4% (in stark contrast to an assisted delivery rate of 0.1% in our study).

Another point to consider when reporting CDRs, is that if the District Health System model is to be reflected by the data, the denominator used to describe delivery rates in this

study should include all live births occurring in primary care facilities which refer to the district hospital. The adjusted CDR which includes live births conducted within the clinics draining to the hospital in 2018 is 22% (Dr Kelly Gate, personal communication).

Moving onto monitoring practices, it was disappointing to note that CD auditing had occurred in only 61.0% of cases sampled. The auditing process used by the facility to determine if CDs were indicated or not was prone to bias (auditing done in an open forum, with the final subjective judgement being made by a non-specialist)–this is evident in the fact that all CDs in the sample had been assessed as 'indicated'. Even if bias were addressed, Allanson et al. [27] note that quality-of-care audits do not necessarily improve outcomes.

WHO CD safety checklist completeness was better (87.7%), but any comment about the fidelity of this process (specifically its impact on improving safety) should be reserved for studies that compare completeness and outcomes at different sites.

The adolescent delivery in-facility rate of 27.1% is higher than the district's most recently published rate of 20.1% (which includes deliveries at all facilities, including primary care), and validates its persistent presence in the top ten list of highest adolescent delivery rate districts [28]. In addition, the adolescent-specific CDR of 25% is comparable to that found by Govender et al. [29] whose study was performed in a similar setting about five years earlier (27.7%).

This should be interpreted in light of the following:

- 36.4% (71 of 195) of all CDs were Robson group 1 (i.e. nulliparas at term with a cephalic singleton pregnancy)

- 53.4% of nulliparas who delivered at the hospital were adolescents

Such findings draw attention to the fact that adolescent nulliparas are a group to focus on, along the continuum of reproductive health, from family planning, through to decision-making around MOD, which would contribute to preventing unnecessary primary CD (especially given that our study suggests a repeat rate of 64.6% after first CD.

Decision-making was not assessed beyond timing. Further research is needed to describe and analyse the complex interplay of human behaviours and attitudes that result in a decision to proceed with a particular MOD. This is also true of anaesthesia, wherein the choice between spinal and general anaesthesia may depend on the knowledge and attitudes of the attending healthcare provider.

## Obstetric outcomes

Evidence involving more than 3500 patients highlights that CDs conducted in the African continent are associated with a 50-fold higher risk of maternal mortality than those conducted in high-income countries [30]. Although no deaths were observed in our study, we found a significant association between CD and maternal complications (largely related to bleeding risk and infection). There was also a seemingly high rate of anaesthetic complications (15.9% of CDs). Mothers delivering by CD had odds three times higher of experiencing a complication, as compared to NVD.

One feature of the study site that is common to South African district hospitals is the lack of specialist staff. It is noted by Aubrey-Bassler et al. [31] that the rate of morbidity at CD is higher among patients managed by generalists as opposed to specialists.

Upskilling is important in cases when patients cannot be transferred out to a higher level of care. A small percentage (3.2%) of parturients in our study remained at district-level, whilst their condition warranted a higher level of expertise. This sub-group's outcomes were not analysed.

In addition to the lack of specialist support within the facility (and sometimes within the district), to assist non-specialists with decision-making around high-risk cases, the 7th Saving Mothers (2014–2016) [7] and 10th Saving Babies (2014–2016) [8] reports list some issues specific to rural and/or primary-level facilities:

- A higher CFR related to CD in rural vs. urban provinces

- A higher institutional maternal mortality ratio (iMMR) for CD compared to vaginal delivery at primary-level facilities

- A higher iMMR related to anaesthesia in rural vs. urban provinces

- A failure in the referral system, namely not referring to the next level of care timeously, and delays in inter-facility emergency transport

- A larger proportion of avoidable factors related to healthcare workers at district-level facilities compared to higher levels of care

The rate of successful vaginal birth after one CD of 58% does not compare favourably with the anticipated success rate of vaginal birth after CD of 72–75% suggested by the Royal College of Obstetricians and Gynaecologists [32]. However, clinical conditions and resources in South Africa differ to those in the United Kindgom. The rates of 33.3% found by Daihoum and Sebitloane [33] and 35% found by Mokaya [34], reflect practices and outcomes at South African urban tertiary facilities, where the expected clinical risk characteristics may predict a lower success rate than that expected at a district-level facility.

The authors are not aware of any literature that has scrutinised complication rates based on indications for CD, however, there are a number of studies which reveal a higher rate of maternal (and neonatal) complications related to emergency CD, as compared to elective CD [35, 36].

## Neonatal outcomes

The 'low Apgar' rate 9.2 per 1 000 livebirths suggests that intrapartum problems are being diagnosed too late, when compared to findings by Padayachee and Ballot [37], who found a perinatal asphyxia rate of 4.7 per 1 000 live births, in a tertiary institution in South Africa between 2006 and 2011.

Looking more specifically at rural South African obstetric services, Gaunt [26] found that the neonatal mortality rate for vacuum deliveries (excluding known stillbirths diagnosed before delivery) was 11.9 per 1 000 (higher than the overall institutional neonatal mortality rate of 9.3 per 1 000), but no statistical analysis was performed. Our study found a neonatal mortality rate of 6.2 per 1 000 livebirths, and, as mentioned, no association between MOD and neonatal transfer out/death.

The normal practice at the study site was for well neonates to 'room in' with their mother in the post-natal ward, whereas neonates requiring any medical attention were admitted to nursery. We found that neonates delivered by CD were four times more likely to require admission to nursery than those delivered by NVD.

## Recommendations

It has been noted that targeted interventions for rural healthcare in South Africa have been few, and that issues which particularly effect rural health facilities, such as a lack of human resources for health, drive avoidable and modifiable factors in maternal and child mortality [38]. This study characterises the correlation between maternal/perinatal outcome and

obstetric practice specific to the South African rural district-level context. Through doing this, it is hoped that the study will have the capacity inform future policies and health system interventions, with a 'rural-proof' focus.

## Limitations

It is important to acknowledge the large potential for information bias in this study at many levels:

- Subjective assessments of blood loss (in clinical practice) and what Lucas class a CD was, and whether it was indicated (in clinical audit)

- Poor quality of documentation regarding neonatal outcomes with possible under-recording of adverse events

- Loss to follow-up of patients (transferred out and not followed up, or discharged and presenting with a puerperal or neonatal problem, but with a new file), which may cause under-representation of the complexity of patients being managed

- Inadequate auditing of severe acute maternal/neonatal morbidity using the same framework as employed in auditing of mortality (i.e. avoidable factors such as administrative or related to healthcare professionals)

- The relatively small number of patients in the sample makes detailed and confident analysis about impact of the MOD difficult.

Some of these limitations would be overcome by an electronic health-record, but because such systems do not, in general, exist at rural district-level, this study gives a unique perspective on the realities of obstetric practice in this setting.

## Conclusion

This study highlighted a relatively high in-facility CDR of 30.8% at this deep rural district-level hospital. Although all CDs were deemed to be indicated, as mentioned, this assessment is prone to bias. Most patients were managed at the correct level of care despite the challenges in the referral system inherent in rural practice. With a three-fold higher risk of maternal complications, CD at rural primary care level was validated as a potentially dangerous undertaking, for which adequate precautions should be taken.

## Supporting information

**S1 File. Personal communication from Dr Kelly Gate.**
(PDF)

**S2 File. Research proposal.**
(PDF)

**S1 Dataset.**
(XLSX)

## Acknowledgments

Prof. David Bishop
Dr. Neil Moran
    Prof. Jagedisa Moodley

Dr. Radhika Singh

Ms. Tonya Esterhuizen

Dr. Boikhutso Tlou

## Author Contributions

**Conceptualization:** Adam Konrad Asghar, Thandaza Cyril Nkabinde, Mergan Naidoo.

**Data curation:** Adam Konrad Asghar.

**Formal analysis:** Adam Konrad Asghar.

**Funding acquisition:** Adam Konrad Asghar.

**Investigation:** Adam Konrad Asghar.

**Methodology:** Adam Konrad Asghar, Thandaza Cyril Nkabinde, Mergan Naidoo.

**Project administration:** Adam Konrad Asghar.

**Resources:** Adam Konrad Asghar.

**Supervision:** Thandaza Cyril Nkabinde, Mergan Naidoo.

**Validation:** Adam Konrad Asghar.

**Visualization:** Adam Konrad Asghar.

**Writing – original draft:** Adam Konrad Asghar.

**Writing – review & editing:** Adam Konrad Asghar, Thandaza Cyril Nkabinde, Mergan Naidoo.

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
