## [Decision Letter · Decision Letter 0]

23 Aug 2021

PONE-D-21-22819

An analysis of obstetric practices and outcomes in a deep rural district hospital in South Africa

PLOS ONE

Dear Dr. Asghar

Thank you for submitting your manuscript to PLOS ONE. After careful consideration, we feel that it has merit but does not fully meet PLOS ONE’s publication criteria as it currently stands. Therefore, we invite you to submit a revised version of the manuscript that addresses the points raised during the review process.

We look forward to receiving your revised manuscript.

Kind regards,

Yogan Pillay, Phd

Academic Editor

PLOS ONE

Journal Requirements:

Additional Editor Comments:

It is recommended that the authors explore indications for caesarean deliveries further and compare complication rates.

65% of CD were classified as Lucas II and 26% were elective. Authors should compare maternal and neonatal outcomes for each of the Lucas classes.

Avoidable factors such as administrative/or medical related should also be determined in cases where complications occurred. These have been shown in reports from the National Committee on Confidential Enquiries into Maternal Mortality in South Africa to be substantial causes of institutional maternal mortality.

Reviewers' comments:

Reviewer's Responses to Questions

**Comments to the Author**

1. Is the manuscript technically sound, and do the data support the conclusions?

Reviewer #1: Yes

2. Has the statistical analysis been performed appropriately and rigorously? 

Reviewer #1: Yes

3. Have the authors made all data underlying the findings in their manuscript fully available?

Reviewer #1: Yes

4. Is the manuscript presented in an intelligible fashion and written in standard English?

Reviewer #1: Yes

5. Review Comments to the Author

Reviewer #1: This is an important study which highlights the potential risks associated with caesarean deliveries undertaken in rural settings in South Africa. It is important to identify the cause of the problem so that solutions/recommendations can be made to reduce this risk. I suggest that the authors explore indications for caesarean deliveries further and compare complication rates.

65% of CD were classified as Lucas II and 26% were elective. Authors should compare maternal and neonatal outcomes for each of the Lucas classes.

Avoidable factors such as administrative/or medical related should also be determined in cases where complications occurred.

6. PLOS authors have the option to publish the peer review history of their article (what does this mean?). If published, this will include your full peer review and any attached files.

Reviewer #1: No

---

## [Author Response · Author response to Decision Letter 0]

24 Sep 2021

Comment: I suggest that the authors explore indications for caesarean deliveries further and compare complication rates.

Response: This was an oversight on our part not to look at this in the research objectives. A secondary analysis reveals no statistically significant association between CD indication and complications rates. We have not included the details of this analysis in the manuscript. No literature was found to highlight previously described complication rates relating to CD indication.

Lines: 265-266; 434-437

Comment: Authors should compare maternal and neonatal outcomes for each of the Lucas classes.

Response: Thank you for bringing this point up, which is important. A secondary analysis reveals no statistically significant association between Lucas classes and maternal/complication rates. We have not included the details of this analysis in the manuscript. Most literature compares emergency vs. elective CD, rather than the individual Lucas classes.

Lines: 265-266; 434-437

Comment: Avoidable factors such as administrative/or medicalrelated should also be determined in cases where complications occurred. These have been shown in reports from the National Committee on Confidential Enquiries into Maternal Mortality in South Africa to be substantial causes of institutional maternal mortality.

Response: Unfortunately, data was not collected regarding these, as they were not part of the study’s objectives. This is, however, a pertinent point. From the data sources available, it would have been difficult to ascertain avoidable factors, as cases of maternal and neonatal morbidity were not discussed at facility Perinatal Morbidity and Mortality Meetings in the same depth as mortality (information bias). We acknowledge that severe acute maternal and neonatal morbidity needs to be thoroughly investigated, and have thus included this oversight as a limitation of the study.

Lines: 481-483

---

## [Editor Report · Decision Letter 1]

21 Dec 2021

An analysis of obstetric practices and outcomes in a deep rural district hospital in South Africa

PONE-D-21-22819R1

Dear Dr.Asghar

We’re pleased to inform you that your manuscript has been judged scientifically suitable for publication and will be formally accepted for publication once it meets all outstanding technical requirements.

Kind regards,

Yogan Pillay, Phd

Academic Editor

PLOS ONE
---

## [Editor Report · Acceptance letter]

23 Dec 2021

PONE-D-21-22819R1 

An analysis of obstetric practices and outcomes in a deep rural district hospital in South Africa 

Dear Dr. Asghar:

I'm pleased to inform you that your manuscript has been deemed suitable for publication in PLOS ONE. Congratulations! Your manuscript is now with our production department. 

Kind regards, 

on behalf of

Dr. Yogan Pillay 

Academic Editor

PLOS ONE